# Advance personal planning knowledge, attitudes, and participation amongst community-dwelling older people living in regional New South Wales, Australia: A cross-sectional survey

Emilie C. Cameron[1,2]*, Nola Ries[3], Amy Waller[1], Briony Johnston[3], John Anderson[4], Jamie Bryant[1,2]

1 School of Medicine and Public Health, College of Health, Medicine and Wellbeing, University of Newcastle, Callaghan, NSW, Australia, 2 Equity in Health and Wellbeing Research Program, Hunter Medical Research Institute, New Lambton Heights, NSW, Australia, 3 Faculty of Law, University of Technology Sydney, Sydney, NSW, Australia, 4 School of Law and Justice, College of Human & Social Futures, University of Newcastle, City Campus, Newcastle, NSW, Australia

* emilie.cameron@newcastle.edu.au

## Abstract

### Background

Advance personal planning (APP) involves planning for future periods of incapacity, including making legal decisions and documents. APP ensures that a person's values and preferences are known and respected. This study aimed to examine knowledge of APP, attitudes and confidence towards APP, and participation in APP activities among older people residing in regional and rural areas.

### Methods

A cross-sectional survey was conducted with people aged over 65 years residing in and around regional towns in New South Wales, Australia. Participants responded to a social media advertisement or information provided through a community organisation. Data was collected via pen and paper survey or an online survey. The survey was developed for the study and included questions about the participant and their experiences with APP. Poisson regression modelling was conducted to explore the relationship between APP participation and APP knowledge, confidence and attitudes as well as the participant characteristics associated with APP participation.

### Results

Overall, 216 people completed the survey. Most participants had a will (90%) but only a third (32%) had documented an advance care directive. Knowledge of APP was low with only 2.8% of participants correctly answering all 6 knowledge questions. Participants had a positive attitude towards APP and high level of confidence that they could discuss APP issues

**Data Availability Statement:** The data that support the findings of this study are available on request from the corresponding author, Emilie Cameron, or

by request to the Hunter New England Health Human Research Ethics Committee via HNELHD-ResearchOffice@health.nsw.gov.au. The data are not publicly available due to ethical restrictions on sharing the data that could compromise the privacy of research participants.

**Funding:** This work was supported by an Australian Research Council Discovery Program grant (DP190100861) and infrastructure funding from the Hunter Medical Research Institute. The funders had no role in study design, data collection and analysis, decision to publish, or preparation of the manuscript.

**Competing interests:** The authors have declared that no competing interests exist.

with important people in their life. Those with increased knowledge, confidence and attitude towards APP were significantly more likely to participate in APP activities. Older age and having private health insurance were significantly associated with engaging in APP activities. Increased frailty and the presence of health conditions were not associated with increased APP participation.

## Conclusions

There is a need to increase engagement with APP particularly among those who may be considered frail or have chronic health conditions. Increasing knowledge of, confidence and attitudes towards, APP could help to increase engagement in APP activities.

## Introduction

Advance Personal Planning (APP) is a process that allows an individual to consider, discuss and record their values and preferences about future financial, health and personal matters in anticipation of potential later loss of capacity or death [1, 2]. APP encompasses the full range of available mechanisms for planning ahead for future health, legal, financial, and personal matters [2]. These include: a Will for the beneficial allocation of property and assets following death; the appointment of an Enduring Power of Attorney authorised to make decisions relating to financial matters during a period of incapacity; the appointment of an Enduring Guardian authorised to make decisions regarding health and personal matters if an individual is not able to communicate or make these choices for themselves; and an Advance Care Directive that sets out a person's values and preferences relating to their future medical treatment or health care [3–5]. The process of Advance personal planning also includes ongoing discussions with health and legal professionals and other important people [6, 7].

There is increasing recognition that engaging in APP can be beneficial for older adults, their families and wider society [2, 8]. As people age, they may experience a range of changes such as physical and cognitive vulnerability [9, 10]; changes in financial circumstances [5, 11], family circumstances [12, 13], accommodation [14, 15], and social expectations, participation and needs [16, 17]; and facing the end of life and bereavement [18, 19]. Many older adults (defined as those aged 65 years or over) are also at risk of periodic, temporary or permanent loss of capacity, which can impact their ability to manage their affairs in a way that is consistent with their self-interest and values [11, 20–22]. Having APP documents in place before they are needed helps to ensure that decisions are made in line with a person's values and preferences [2, 8]. Knowing who is responsible for making these decisions through the appointment of substitute decision makers, including an Enduring Power of Attorney for decisions about property and finances and an Enduring Guardian for personal and medical care, helps to reduce stress and avoid disputes. Having a person's wishes documented in an Advance Care Directive can help to give the decision maker confidence in their decisions and make them aware of the person's preferences so they can make fully informed decisions. A will comes into place after death and documents the distribution of a person's possessions and assets [2, 23].

Much of the previous work done on APP focuses on prevalence of advance care planning, or planning for future medical care, rather than broader forms of APP among older adults. For instance, a recent national study [24] found an overall advance care planning documentation prevalence of 46% and Advance Care Directive prevalence of 25% among 4187 older Australians attending 51 health and residential aged care services. Predictors of advance care

planning documentation included being female, older, having two or more medical conditions, receiving palliative care, being divorced/separated, and living in a residential aged care facility (RACF). Having discussed advance care planning with someone else has been reported as a strong predictive factor for completing some form of advance care planning documentation [25], however, older adults report avoiding these types of discussions due to a fear and reluctance to think about dying and end-of-life [26]. Older adults also report limited awareness regarding who is responsible for initiating advance care planning conversations [27]; and limited knowledge and confidence in their ability to carry out advance care planning [27, 28]. Much of the previous work represents a siloed account of advance planning as a purely medical or legal problem rather than a holistic APP process that considers financial, health and personal matters.

Integrating APP as part of usual health care and legal services is especially relevant for older adults residing in regional and rural areas as they often experience a number of unique issues including intergenerational succession of family businesses [4, 29]. Rates of health disadvantage are higher in regional areas, which contributes to increased hospitalisation, a greater range of risk factors and less access to health services [30, 31]. Further distances to travel and fewer transportation options also contribute to difficulties with accessing health care [32, 33], and can also impact access to legal services, employment and financial resources, social support and opportunities for social interactions [34, 35]. Despite the unique needs of older people residing in regional and rural areas, there is limited research about engagement in APP activities amongst this population group. Understanding how older adults in regional and rural areas are engaging with the full range of available APP mechanisms will support strategies to promote APP in this population, and help prevent legal, health, financial and personal conflicts arising in future. Therefore, the aims of this study were to examine in a sample of community-dwelling older people residing in regional or rural areas of New South Wales (NSW), Australia, their self-reported:

1. Knowledge of APP activities;

2. Perceived benefits and risks in engaging in APP activities;

3. Confidence in their ability to discuss and document their wishes; and

4. Participation in APP activities and the factors associated with increased participation.

## Methods

### Design

A descriptive cross-sectional survey.

### Participants

Ten NSW towns were randomly selected for participation based on their population size (between 5,000 and 20,000) and age distribution (at least 15% of population aged over 65). Eligible participants were aged 65 years or over, residing in or around the selected regional or rural NSW town, and able to read and understand English.

### Recruitment and data collection

Potential participants were recruited via two methods.

1. Community organisations frequented by older adults in the selected regional towns were identified by the research team, and an invitation letter mailed to their President (or similar) requesting their support for the study. A follow-up telephone call occurred after 7 days to ensure the invitation was received and directed to the appropriate contact person. Consenting community organisations identified potentially eligible members on their membership lists and mailed or handed out Study Recruitment Packages containing a cover letter, information statement, consent form and pen and paper survey to eligible members. Consenting participants returned the completed survey with signed written consent form directly to the research team. Community organisations were contacted and approached members between 1st March and 30th September 2021.

2. Paid Facebook advertisements were run over a 3-week period in September 2021 and March 2022. The advertisements were targeted to people aged 65 and over residing in and around the selected towns. A selection of positive ageing images were accompanied by text stating that the study was about planning ahead, involved completing a survey, and that those who completed the survey would be entered into a draw for a $100 gift voucher. Facebook advertisements have the advantage of reaching people who may be difficult to reach by other methods [36]. Those viewing the ads who were interested in learning more were directed to a webpage with information about the study, the information statement and a secure link to provide consent and contact details. Consenting participants could then complete the survey online or request a paper copy be mailed to them.

The eligibility of participants was checked by collecting their date of birth and postcode. A selection of participants were called to verify their details. All eligible participants were entered into a draw to receive a $100 gift voucher.

## Measures

The survey was developed for this study. Survey items were derived from previous studies undertaken by the authors [37, 38], and from published surveys (e.g. the Advance Care Planning Engagement Survey [39]), then reviewed by a panel including geriatricians, general practitioners, lawyers, physicians and behavioural scientists until consensus on content and format of items was reached. Items were then modified and pilot tested with five older people for acceptability, relevance and clarity, and refined based on their feedback. Scoring of items was based on the Advance Care Planning Engagement Survey which demonstrates good reliability and validity [39].

**APP knowledge.** Six statements developed for the study assessed knowledge of APP. Participants could respond true, false or unsure.

**APP confidence and attitudes.** A modified version of the Advance Care Planning Engagement Survey [39] was used to assess self-efficacy or confidence (7 items) and attitudes towards APP (4 items). The survey measures multiple APP behaviours that relate to engaging decision makers, considering acceptable quality of life and having conversations with key others (e.g. doctors). Items were scored on a 5-point Likert scale from strongly agree to strongly disagree.

**Self-reported APP.** Participants indicated whether they had completed financial APP instruments including a will and Enduring Power of Attorney, and medical instruments including an Advance Care Directive and Enduring Guardian ('yes', 'no', or 'unsure' for each item). Standard, easy to understand definitions were provided. Participants were also asked to indicate if they had discussed their wishes for care with families or other important people in their lives; health care professionals; or lawyers.

**Participant characteristics.**   Participants self-reported their age, sex, marital status, living arrangements, Aboriginal and Torres Strait Islander status, country of birth, highest level of education, employment status, health insurance, whether they received a home care package, and the presence of long term health conditions.

**Frailty.**   Frailty was assessed using the 15-item Tilburg Frailty Indicator [40] which covers: physical, psychological and social components of frailty. Frailty is associated with an increased risk of adverse outcomes in community-dwelling older people, including falls, delirium, disability, residential care admission, hospitalisation and mortality.

**Health and legal service use.**   Service use was assessed by asking participants to report the number of hospital admissions, emergency department (ED) presentations, visits to general practice, consultations with a lawyer/solicitor, and consultations with a financial planner in the last 12 months.

## Statistical analysis

Analyses were conducted in SAS v9.4 (SAS Institute, Cary, North Carolina, USA). Characteristics of participants are reported as means and standard deviations for continuous variables and number and proportion for categorical variables. Aggregated responses to knowledge, confidence, attitudes, and participation are presented as well as knowledge and APP participation scores and mean confidence and attitude to APP. The knowledge score was calculated by summing the number of items answered correctly. Missing and unsure responses were treated as incorrect (1% of responses missing). A composite APP participation outcome on a scale from 0–7 was created by counting the number of APP activities completed. Missing responses (1%) were taken as not completed. A mean confidence score was calculated by averaging the five-point Likert scores across the confidence items, with a lower number indicating greater agreement. A mean attitude score was calculated in a similar way. For this, the item "is not needed because my family or others close to me know my wishes already" was reversed due to the question being posed in the negative. The proportion of missing observations for both confidence and attitude scores was 3%. Four or more of the items needed to be answered for the mean score to be calculated.

Poisson regression modelling was conducted to explore demographic and health factors (age, sex, education, country of birth, employment, private health insurance, living arrangements, chronic health conditions and frailty) associated with APP participation. Crude (univariate) models and an adjusted (multivariate) model were explored. Due to the proportion of missing values exceeding 5% for all demographic and health characteristics, a sensitivity analysis with multiple imputation using fully conditional specification was performed using 20 datasets. Poisson regression models were also used to explore the relationship between APP participation and APP knowledge, confidence and attitudes. To provide meaningful interpretations, confidence and attitude scores were scaled by a factor of 10. Interpretation is therefore due to a change of 0.1 units for these factors.

## Ethics approval

This project received ethics approval from the University of Newcastle Human Research Ethics Committee (H-2019-0139). Participants provided written or electronic consent to participate.

## Results

### Sample

A total of 216 people aged 65 years and older completed the survey. Community organisations were sent 502 survey packs of which 88 were returned completed (18% response rate).

Targeted Facebook ads reached a total of 23,500 people of which 123 were eligible and completed the survey.

Characteristics of the sample are shown in Table 1. The mean age of participants was 73.4 years (SD = 6.0). The majority were female (73%, n = 145), born in Australia (87%; n = 173) and married or living with a partner (56%; n = 114). Seventy four percent (n = 151) had a long-term health condition and 29% (n = 61) were considered frail on the Tilburg Frailty Indicator. Almost all participants had visited a GP in the last 12 months (99%, n = 204), while a third had consulted either a lawyer (30%, n = 62) or financial planner (31%, n = 64).

## Knowledge of APP

Table 2 presents the responses given for each APP knowledge item. Only 2.8% (n = 6) of participants gave the correct response for all six knowledge items. The mean knowledge score (i.e. number of correct responses) was 2.9 (standard deviation = 1.5). Three quarters (74%, n = 156) of participants identified that it was false that once wishes for care have been written down they can't be changed. The percentage of participants who responded incorrectly was greatest for the item *An Enduring Power of Attorney can make health care decisions* (58%; n = 124). For each item over 18% were unsure of the answer.

## Attitudes towards APP

Fig 1 presents the proportion of participants who agreed with statements presenting the benefits of APP. Most participants agreed or strongly agreed that APP "makes it easier for my loved ones to make decisions on my behalf" (93%), "gives a better chance of getting the health care I want" (88%) and "helps make sure money and assets are managed the way I want" (86%). However, over a third (33%) thought that APP was "not needed because my family or others close to me know my wishes already". The mean attitude score was 2.01 (SD = 0.54).

## Confidence participating in APP

The proportion of participants who felt confident they would talk with important people in their life about APP and their wishes for future care is shown in Fig 2. Most participants (68%) agreed or strongly agreed with all items (mean = 1.65, SD = 0.54).

## Participation in APP activities

Overall, 90% (n = 192) of participants had made a will, 67% (n = 143) had appointed an Enduring Power of Attorney, 54% (n = 115) had appointed an Enduring Guardian and 32% (n = 67) had made an Advance Care Directive. Discussions had occurred more frequently with family (n = 132, 62%) than lawyers (n = 71, 33%) or health professionals (n = 42, 20%). Fig 3 presents the number of APP activities participants self-reported engaging in. Overall, 9.3% (n = 20) of the respondents had engaged in all seven APP activities and 5.1% (n = 11) had not engaged in any of the identified APP activities.

## Characteristics of those who engaged in APP

Table 3 shows the results of multiple logistic regression modelling. When adjusted for other factors, age was significantly associated with increased APP participation at the 5% level. For every 1 year increase in age there was a 3% increase in the rate of participation in APP activities. Similarly, those with private health insurance had a 23% increase in the rate of participation in APP activities compared to those without. Those who had a chronic health condition showed a 17% decreased rate of participation in APP activities compared to those who didn't

**Table 1. Demographics of the sample (n = 216).**

| | | Mean | SD |
|---|---|---|---|
| **Age** | years | 73.4 | 6.0 |
| | | **n** | **%** |
| **Sex** | Male | 54 | 27% |
| | Female | 145 | 73% |
| | Missing | 17 | |
| **Highest education** | Primary School/High School | 68 | 34% |
| | Trade or vocational education | 54 | 27% |
| | University degree | 80 | 40% |
| | Missing | 14 | |
| **Country of birth** | Australia | 173 | 87% |
| | Other-UK | 15 | 8% |
| | Other | 12 | 6% |
| | Missing | 16 | |
| **Aboriginal and Torres Strait Islander** | Yes | 5 | 2.5% |
| | No | 194 | 97% |
| | Missing | 17 | |
| **Employment status** | Full time or part time work | 34 | 17% |
| | No paid work (retired/ pension/ home duties/ volunteer work) | 167 | 83% |
| | Missing | 15 | |
| **Marital status** | Married or living with partner | 114 | 56% |
| | Divorced | 38 | 19% |
| | Widowed | 42 | 21% |
| | Never married | 8 | 4% |
| | Missing | 14 | |
| **Living arrangements** | Spouse or partner and/or children | 114 | 57% |
| | Other family members or friend | 7 | 3% |
| | Alone | 82 | 40% |
| | Missing | 13 | |
| **Health insurance** | Yes | 137 | 67% |
| | No | 66 | 33% |
| | Missing | 13 | |
| **Concession card** | Yes | 155 | 78% |
| | No | 45 | 23% |
| | Missing | 16 | |
| **Health conditions** | No long term health conditions | 53 | 26% |
| | At least 1 long term health condition | 151 | 74% |
| | Missing | 12 | |
| **Frailty** | Not frail (0–4) | 153 | 71% |
| | Frail (5–12) | 61 | 29% |
| | Missing | 2 | |
| **Receive home care package** | Yes or on waiting list | 22 | 11% |
| | No | 186 | 89% |
| | Missing | 8 | |
| **Service use in past 12 months** | Hospital admission | 55 | 26% |
| | Emergency department | 47 | 23% |
| | General practitioner | 204 | 99% |
| | Lawyer/solicitor | 62 | 30% |
| | Financial planner | 64 | 31% |
| | Missing | 8–9 | |

**Table 2. Self-reported knowledge about APP.** Correct responses are indicated in bold (n = 210–214).

| To the best of your knowledge: | True | False | Unsure |
|---|---|---|---|
| An Enduring Power of Attorney can make health care decisions | 124 (58%) | **49 (23%)** | 41 (19%) |
| An Advance Care Directive must be witnessed by a lawyer | 76 (36%) | **45 (21%)** | 93 (43%) |
| Once a person has written down their wishes for care (e.g. in an advance care directive), they can't change it | 14 (6.7%) | **156 (74%)** | 40 (19%) |
| A person may have more than one Enduring Guardian at a time | **106 (50%)** | 28 (13%) | 80 (37%) |
| An Enduring Guardian can make decisions for a person who is still able to make decisions for themselves | 26 (12%) | **140 (65%)** | 48 (22%) |
| A properly made Advance Care Directive that refuses treatment should be followed, even if the doctor disagrees with it | **130 (61%)** | 15 (7.0%) | 69 (32%) |

have a chronic health condition, although this was not significant in a sensitivity analysis using multiple imputation for missing values. The sensitivity analysis did not change the result for any other variable. Sex, education, employment, living arrangements, country of birth and frailty did not show a significant association with engaging in APP.

In univariate Poisson regression models those with a higher knowledge score, more confidence with discussing APP and a more positive attitude towards APP were significantly more likely to engage in APP activities (Table 4).

## Discussion

This study provides insights on attitudes toward and engagement with APP among older people living in rural and regional communities in NSW. Across Australia, regional and remote communities are experiencing greater chronic illness and poorer health outcomes than in urban areas [30]. This study found that most participants reported having a will (90%) and more people reported having appointed an Enduring Power of Attorney (67%) than an Enduring Guardian (54%). Only one third had made an Advance Care Directive (32%). These rates of engagement are nearly identical to the findings of a multi-state study of older Australians (mostly aged 70+) receiving home care services, of whom 60% resided in regional areas [37].

The relatively low rate of Advance Care Directive completion among this older cohort is somewhat surprising, especially since nearly all participants (99%) reported seeing a GP in the previous 12 months. Recent Australian research indicates that GPs, especially those in regional and rural areas, have positive attitudes toward discussing advance care planning with older patients as part of comprehensive 75+ health assessments, and report providing written material about planning and assistance with planning for those who were interested [41]. However,

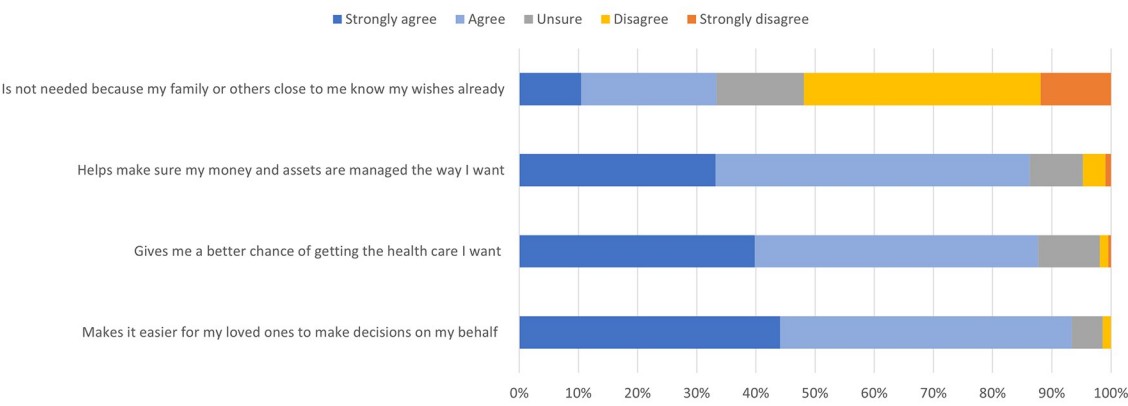

**Fig 1. Self-reported benefits of APP (n = 209–210).**

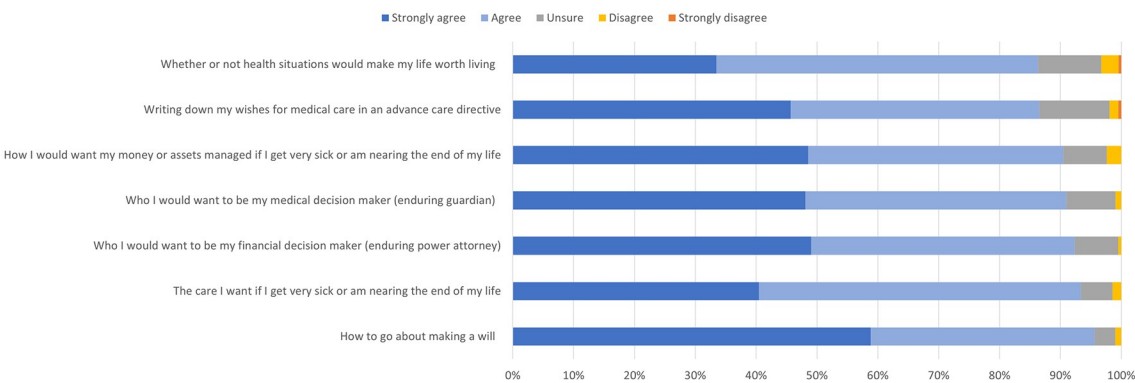

**Fig 2. Proportion of participants agreeing or disagreeing that they feel confident they could talk with others about each APP item (n = 204–211).**

social and personal reluctance to discuss the dying process can stop patients from initiating these discussions [7]. In addition, older people in regional communities may have lower interest in health-related planning if they have long-term relationships with doctors who have practiced in the community for many years. Individuals may see less need for advance care planning if they believe local health practitioners already know their values and preferences for care [42]. However, health workforce demographics are changing in regional areas with increasing use of short term and locum practitioners who do not have sustained connections with the local community [43]. This shift calls for greater emphasis on supporting older people in regional settings to document their wishes for future care and providing multiple avenues to engage with APP.

Most participants (62%) reported they had discussed their wishes for care with family members or other important people in their lives. Such discussions are a vital component of APP to ensure that others know and can advocate for the wishes of the older person in the event of future incapacity [6]. By contrast, discussions were far less commonly reported with lawyers (33%) or health professionals (20%). The lack of discussion with professionals has two concerning implications. First, while many older people said they had wills and Enduring Power

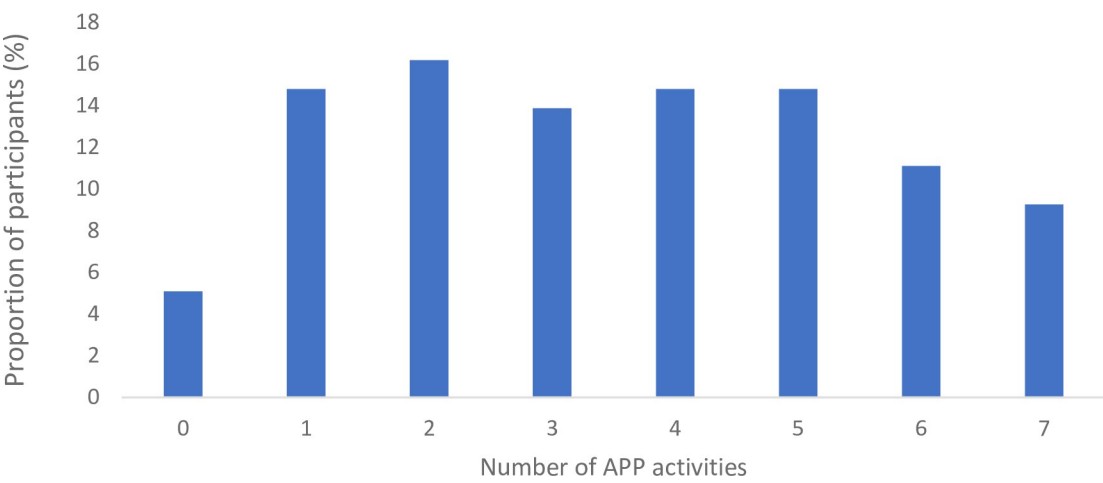

**Fig 3. The total number of advance care planning activities engaged in by participants (n = 211–215).**

**Table 3. Socio-demographic characteristics associated with engaging in advance personal planning activities (n = 188 in adjusted model).**

| | | Crude | | Adjusted | |
|---|---|---|---|---|---|
| | | RR (95%CI) | P-value | RR (95%CI) | P-value |
| **Age** | Continuous | 1.02 (1.01, 1.04) | < .0001 | 1.03 (1.01, 1.04) | < .0001 |
| **Sex** | Male | Reference | | Reference | |
| | Female | 1.01 (0.86–1.20) | 0.87 | 1.04 (0.87–1.26) | 0.66 |
| **Highest education** | School | Reference | | Reference | |
| | Trade or vocational education | 0.97 (0.81, 1.17) | 0.75 | 1.02 (0.84, 1.27) | 0.76 |
| | University degree | 0.92 (0.77, 1.09) | 0.32 | 0.90 (0.75, 1.09) | 0.28 |
| **Country of birth** | Australia | Reference | | Reference | |
| | Other | 0.92 (0.74, 1.15) | 0.45 | 0.91 (0.72, 1.13) | 0.38 |
| **Employment status** | No paid work | Reference | | Reference | |
| | Full time or part time work | 0.94 (0.77, 1.15) | 0.56 | 0.99 (0.79, 1.23) | 0.91 |
| **Living arrangements** | With others | Reference | | Reference | |
| | On own | 1.08 (0.93, 1.25) | 0.29 | 1.10 (0.93, 1.29) | 0.27 |
| **Health insurance** | No | Reference | | Reference | |
| | Yes | 1.22 (1.03, 1.43) | **0.01** | 1.23 (1.03, 1.47) | **0.02** |
| **Health conditions** | No long term health conditions | Reference | | Reference | |
| | At least 1 health condition | 0.89 (0.76, 1.05) | 0.26 | 0.83 (0.70, 0.99) | **0.04** |
| **Frailty** | Not frail (0–4) | Reference | | Reference | |
| | Frail (5–12) | 0.91 (0.78, 1.07) | 0.26 | 0.97 (0.80, 1.18) | 0.75 |

of Attorneys in place, the low rate of recent consultations (30%) with a lawyer suggests these legal instruments were either made in prior years and could be outdated or were completed without consultation with a legal professional. This increases the chance that the will may not reflect current wishes, and/or may be improperly made and vulnerable to future legal challenges. Second, while GPs have an integral role to play in both prompting and facilitating advance care planning, this is not occurring with many patients. Nearly all (99%) respondents had seen a GP in the past 12 months, yet few had had conversations about care planning and those with chronic health conditions or increased frailty did not report increased engagement with APP. Evidence suggests that people with poorer health outcomes positively approach discussions regarding advance care planning, compared to those in good health [44, 45] and that people expect these conversations following a diagnosis [38]. However, GPs operate in a time-limited environment and many report that it can be hard to find the right time to discuss advance care planning amid changing patient needs and preferences [46, 47]. There is also uncertainty from patients and GPs about the process and legal aspects of planning [7, 47, 48]. Improved awareness of GP-focused structured conversation guides (for example www.theadvanceproject.com.au) as well as sources of legal information and referral networks within communities would help to address these issues.

This study revealed several concerning deficiencies in knowledge of older adults about the legal aspects of APP instruments. Other studies have found a similar low level of knowledge about

**Table 4. Association between knowledge, confidence and attitude to APP and participation in APP activities.**

| Score | n | Rate Ratio (95% CI) | P-value |
|---|---|---|---|
| Knowledge score | 216 | 1.11 (1.06, 1.17) | <**0.001** |
| Attitude mean x10 | 210 | 0.97 (0.95, 0.98) | <**0.001** |
| Confidence mean x10 | 209 | 0.96 (0.94, 0.97) | <**0.001** |

APP across different population groups [28, 49–51]. Literature also highlights the risks that may arise from misunderstanding APP instruments and the need for enhancing knowledge to ensure APP achieves its intended purposes [52]. Over three-quarters of respondents had either an incorrect or uncertain understanding about the scope of an Enduring Power of Attorney's legal authority, with 58% wrongly thinking that a person in this role has the authority to make healthcare decisions. This misperception might also contribute to the lower uptake of health-related APP instruments. One-third of respondents are at risk of an Enduring Guardian prematurely taking over healthcare decision-making, due to the lack of understanding that an Enduring Guardian is only meant to act in circumstances where the individual lacks the capacity to make their own decisions. For around one-quarter of respondents, misperceptions that an Advance Care Directive cannot be changed may be a deterrent to making one in the first place. Despite these knowledge gaps, the study provides evidence of strong community sentiment that APP has important benefits, particularly in easing future decision-making burdens for loved ones and ensuring that decisions about healthcare and financial management align with the individual's wishes.

A recent systematic review [53] found that psychological and extrinsic factors were barriers to the uptake of advance care planning. This included having a fear of death, feeling unprepared or not ready for discussions, lacking confidence that their wishes would be followed, uncertainty of when to have APP conversations, a lack of family support, and a feeling that those close to them knew their wishes already. Similar to other studies [50, 51], the current study found that greater knowledge, more positive attitudes and higher levels of confidence in discussing APP were associated with increased participation in APP activities. This suggests that resources to increase these factors in the community could help overcome some of the barriers to APP and have a positive effect on APP engagement. Future work should investigate the effectiveness of such strategies.

It is notable that 40% of the older adults in this study reported living alone. National data indicate that around one quarter of people in Australia over age 65 live alone, however this proportion increases with age [15]. Living alone is a risk factor for social isolation [54] and some individuals in this cohort may not have people they trust to appoint as an Enduring Guardian or Enduring Power of Attorney [55]. Not having a trusted person to appoint as an enduring representative is a known barrier to engaging in advance planning activities and older adults in this situation are more likely to report poorer physical and mental health [56]. Applications for public guardianship may be required in some circumstances for seriously ill older people, however, recent research in a NSW geriatric inpatient setting identified significant delays and deficiencies in these legal processes [57].

### Limitations

This study reports the results of a cross-sectional survey of people from regional and rural areas of NSW. The sample included more people with a university degree than expected in the general population suggesting that the sample may not be representative. Older people are more likely to report that they have a will and enduring financial power of attorney, but this study did not investigate whether these legal documents are up-to-date and validly formulated. Further, the survey did not explore issues related to support for decision-making.

### Conclusions

Strategies are needed to help older people living in regional areas engage with the full range of APP activities, especially those experiencing frailty and progressive health conditions. Increasing community knowledge and attitudes towards APP could help, as well as health and legal professionals promoting a proactive approach to APP in this population.

## Acknowledgments

The authors thank Sandra Dowley and Sarah Leask for assistance with data entry and analysis and Wendell Peacock and all community organisations who assisted with recruitment.

## Author Contributions

**Conceptualization:** Nola Ries, Amy Waller, John Anderson.

**Data curation:** Emilie C. Cameron.

**Formal analysis:** Emilie C. Cameron.

**Funding acquisition:** Nola Ries, Amy Waller, John Anderson, Jamie Bryant.

**Methodology:** Emilie C. Cameron, Nola Ries, Briony Johnston.

**Project administration:** Nola Ries, Amy Waller, Jamie Bryant.

**Writing – original draft:** Emilie C. Cameron, Amy Waller, Briony Johnston.

**Writing – review & editing:** Emilie C. Cameron, Nola Ries, Briony Johnston, John Anderson, Jamie Bryant.

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
