## [Decision Letter · Decision Letter 0]

16 May 2024

PONE-D-23-38030Advance personal planning knowledge, attitudes, and participation amongst community-dwelling older people living in regional New South Wales, Australia: a cross-sectional surveyPLOS ONE

Dear Dr. Cameron,

Thank you for submitting your manuscript to PLOS ONE. After careful consideration, we feel that it has merit but does not fully meet PLOS ONE’s publication criteria as it currently stands. Therefore, we invite you to submit a revised version of the manuscript that addresses the points raised during the review process.

The manuscript has received an overall positive feedback from the reviewer, but warrants few additional refinements. You are requested to please provide the revised manuscript as per the comments below, for further processing and consideration.==============================

We look forward to receiving your revised manuscript.

Kind regards,

Yogesh Kumar Jain, MPH

Academic Editor

PLOS ONE

Journal Requirements:

2. In the online submission form, you indicated that the data that support the findings of this study are available on request from the corresponding author, Emilie Cameron, or by request to the Hunter New England Health Human Research Ethics Committee via HNELHD-ResearchOffice@health.nsw.gov.au. The data are not publicly available due to ethical restrictions on sharing the data that could compromise the privacy of research participants.

Additional Editor Comments:

Dear Authors,

The manuscript has received an overall positive feedback from the reviewer, but warrants few additional refinements, such as explanations of EPOA, ACD, and EG; clarification of selection process; information about the reliability and scoring of study tools; and approach to handling missing data.

You are requested to please provide in line explanations and justification of omissions wherever necessary for further processing.

Reviewers' comments:

Reviewer's Responses to Questions

**Comments to the Author**

1. Is the manuscript technically sound, and do the data support the conclusions?

Reviewer #1: Partly

2. Has the statistical analysis been performed appropriately and rigorously? 

Reviewer #1: Yes

3. Have the authors made all data underlying the findings in their manuscript fully available?

Reviewer #1: No

4. Is the manuscript presented in an intelligible fashion and written in standard English?

Reviewer #1: Yes

5. Review Comments to the Author

Reviewer #1: Thank you, editor, for inviting me to review the paper. The paper makes a good impression at the beginning but requires improvements in other parts of the manuscript. The following are my suggestions for the improvement of this manuscript:

Abstract: Please consider including the type of data analysis involved and the questionnaires for data collection.

Introduction: I recommend providing additional explanations about EPOA, ACD, and EG, including their benefits, disadvantages, and differences.

Methodology: the authors should clarify how the random selection process was performed. The statement” organisations identified potentially eligible members 119 on their membership list”, does not align with random sampling principles. For Facebook advertisements, how do you ensure that only older people residing in and around selected towns respond to the surveys? Additionally, please include information about the reliability and scoring of the study instruments. Lastly, describe your approach to handling missing data.

Results & Discussion: Please provide examples of countries for the 'other' category in the demographic data. Avoid using too many abbreviations like EPOA, ACD, and EG as they can disrupt the flow of the manuscript; consider using the full terms instead. For the statement ‘This study revealed several concerning deficiencies in knowledge of older adults about the legal 327 aspects of APP instruments”, please elaborate on how this compares to previous findings. Please include recommendations for future studies.

6. PLOS authors have the option to publish the peer review history of their article (what does this mean?). If published, this will include your full peer review and any attached files.

Reviewer #1: No

---

## [Author Response · Author response to Decision Letter 0]

3 Jun 2024

Thank you for providing reviewer comments on our manuscript. We appreciate the insights provided by the reviewer and have worked to address all suggestions made. Our responses to the comments are provided below and have been made in the revised manuscript.

Editor’s comments

1. Please ensure that your manuscript meets PLOS ONE's style requirements. 

We have checked the document and are confident that it meets the requirements. 

2. In the online submission form, you indicated that the data that support the findings of this study are available on request from the corresponding author, …. The data are not publicly available due to ethical restrictions on sharing the data that could compromise the privacy of research participants.

Making the data public at this point would breach compliance with the approved protocol by the research ethics board. In the ethical approval to this study it states that: “De-identified data may be made available for secondary analysis however separate ethics approval will be sought beforehand. Where data is used for further analysis, it will not contain any identifying information. Only grouped data will be presented in any reports of publications arising from this research.” This is what has been communicated to participants in the Participant Information statement. We therefore request that an exemption be made and the original statement on data availability stands. 

2. Please review your reference list to ensure that it is complete and correct.

We have checked the reference list. 

Reviewer comments: 

• Abstract: Please consider including the type of data analysis involved and the questionnaires for data collection.

We have added a description of the questionnaires used and data analysis performed to the abstract. (page 2 line 30)

“The survey was developed for the study based on previous work. It included questions about the participant and their experiences with APP including their participation, knowledge, confidence and attitude. Poisson regression modelling was conducted to explore the relationship between APP participation and APP knowledge, confidence and attitudes as well as the participant characteristics associated with APP participation.”

• Introduction: I recommend providing additional explanations about EPOA, ACD, and EG, including their benefits, disadvantages, and differences.

We have added further explanation about the documents in the introduction (Page 4 line 68).

“Having APP documents in place before they are needed helps to ensure that decisions are made in line with a person’s values and preferences [2]. Knowing who is responsible for making these decisions through the appointment of substitute decision makers, including an Enduring Power of Attorney for decisions about property and finances and an Enduring Guardian for personal and medical care, helps to reduce stress and avoid disputes. Having a person’s wishes documented in an Advance Care Directive can help to give the decision maker confidence in their decisions and make them aware of the person’s preferences so they can make fully informed decisions. A will comes into place after death and documents the distribution of a person’s possessions and assets. [2, 23]”

• Methodology: the authors should clarify how the random selection process was performed. The statement” organisations identified potentially eligible members 119 on their membership list”, does not align with random sampling principles.

As this was a cross sectional survey the aim is not to randomly select participants but to get a representative sample. Participants were selected from a number of different regions and organisations and using different methods to try to ensure this.

• Methodology: For Facebook advertisements, how do you ensure that only older people residing in and around selected towns respond to the surveys? 

Facebook advertisements were set to only display to the specific population group that we were targeting. All participants provided their age and postcode at the beginning of the survey. They were later asked for their date of birth and suburb as a check of the information provided. The details of selected participants were also verified through a phone call. We have added further details to the methods (Page 8 line 149).

“The eligibility of participants was checked by collecting their date of birth and postcode. A selection of participants were called to verify their details.”

• Methodology: Additionally, please include information about the reliability and scoring of the study instruments.

We have clarified that the survey was developed for the study (Methods page 8 line 153). We have also added further details about the instrument on which it is based (Page 8 line 158).

“Scoring of items was based on the Advance Care Planning Engagement Survey which demonstrates good reliability and validity [41].”

• Methodology: Lastly, describe your approach to handling missing data.

On page 10 of the methods we have stated for each score how missing values were handled. A change has been made to help clarify this. For some demographic variables the proportion of missing values was greater than 5% therefore a sensitivity analysis (as explained on page 10 line 211) was conducted. We have added more to explain this in the results section (page 15 line 286). 

“The sensitivity analysis did not change the result for any other variable.”

• Results & Discussion: Please provide examples of countries for the 'other' category in the demographic data. 

We have added the largest “Other” category (UK) to the demographics table (page 12, Table 1). All other countries had just 1 person from them.

• Results & Discussion: Avoid using too many abbreviations like EPOA, ACD, and EG as they can disrupt the flow of the manuscript; consider using the full terms instead. 

We have removed many of the abbreviations throughout the manuscript using the full version instead, including EG, EPOA, ACD and ACP.

• Results & Discussion: For the statement ‘This study revealed several concerning deficiencies in knowledge of older adults about the legal 327 aspects of APP instruments”, please elaborate on how this compares to previous findings. 

We have added a sentence and references highlighting the results of other studies (Page 18 line 351)

“Other studies have found a similar low level of knowledge about APP across different population groups [51-54]” Literature also highlights the risks that may arise from misunderstanding APP instruments and the need for enhancing knowledge to ensure APP achieves its intended purposes [55].”

• Results & Discussion: Please include recommendations for future studies.

We have included a statement in the discussion on future directions. (page 19 line 373)

“Future work should investigate the effectiveness of such strategies.”

We hope these responses meet with your approval and look forward to receiving your response.

---

## [Decision Letter · Decision Letter 1]

7 Aug 2024

Advance personal planning knowledge, attitudes, and participation amongst community-dwelling older people living in regional New South Wales, Australia: a cross-sectional survey

PONE-D-23-38030R1

Dear Dr. Emilie Cameron

We’re pleased to inform you that your manuscript has been judged scientifically suitable for publication and will be formally accepted for publication once it meets all outstanding technical requirements.

Kind regards,

Shadia Hamoud Alshahrani, PhD

Academic Editor

PLOS ONE

Reviewers' comments:

Reviewer's Responses to Questions

**Comments to the Author**

1. If the authors have adequately addressed your comments raised in a previous round of review and you feel that this manuscript is now acceptable for publication, you may indicate that here to bypass the “Comments to the Author” section, enter your conflict of interest statement in the “Confidential to Editor” section, and submit your "Accept" recommendation.

Reviewer #2: All comments have been addressed

2. Is the manuscript technically sound, and do the data support the conclusions?

Reviewer #2: Yes

3. Has the statistical analysis been performed appropriately and rigorously? 

Reviewer #2: Yes

4. Have the authors made all data underlying the findings in their manuscript fully available?

Reviewer #2: Yes

5. Is the manuscript presented in an intelligible fashion and written in standard English?

Reviewer #2: Yes

6. Review Comments to the Author

Reviewer #2: Thank you for the opportunity to review the manuscript. The manuscript is comprehensively revised. Few amendment to clarify methods:

1. Self-reported APP - how many item?

2. Frailty - scale used?

3. Health and legal service use- how many items?

7. PLOS authors have the option to publish the peer review history of their article (what does this mean?). If published, this will include your full peer review and any attached files.

Reviewer #2: No

---

## [Editor Report · Acceptance letter]

9 Aug 2024

PONE-D-23-38030R1 

PLOS ONE

Dear Dr. Cameron, 

I'm pleased to inform you that your manuscript has been deemed suitable for publication in PLOS ONE. Congratulations! Your manuscript is now being handed over to our production team.

Kind regards, 

on behalf of

Dr. Shadia Hamoud Alshahrani 

Academic Editor

PLOS ONE